# Tracking and Analyzing Public Emotion Evolutions During COVID-19: A Case Study from the Event-Driven Perspective on Microblogs

**DOI:** 10.3390/ijerph17186888

**Published:** 2020-09-21

**Authors:** Qi Li, Cong Wei, Jianning Dang, Lei Cao, Li Liu

**Affiliations:** 1Beijing Key Lab of Applied Experimental Psychology, Faculty of Psychology, Beijing Normal University, Beijing 100875, China; 11112018083@bnu.edu.cn (Q.L.); weicong@bnu.edu.cn (C.W.); jndang@bnu.edu.cn (J.D.); 2Department of Computer Science and Technology, Tsinghua University, Beijing 100085, China; cao-l17@mails.tsinghua.edu.cn

**Keywords:** public emotion, disaster response, microblog, event extraction, sentiment mining

## Abstract

Objective: Coronavirus disease 2019 (COVID-19) has caused substantial panic worldwide since its outbreak in December 2019. This study uses social networks to track the evolution of public emotion during COVID-19 in China and analyzes the root causes of these public emotions from an event-driven perspective. Methods: A dataset was constructed using microblogs (n = 125,672) labeled with COVID-19-related super topics (n = 680) from 40,891 users from 1 December 2019 to 17 February 2020. Based on the skeleton and key change points of COVID-19 extracted from microblogging contents, we tracked the public’s emotional evolution modes (accumulated emotions, emotion covariances, and emotion transitions) by time phase and further extracted the details of dominant social events. Results: Public emotions showed different evolution modes during different phases of COVID-19. Events about the development of COVID-19 remained hot, but generally declined, and public attention shifted to other aspects of the epidemic (e.g., encouragement, support, and treatment). Conclusions: These findings suggest that the public’s feedback on COVID-19 predated official accounts on the microblog platform. There were clear differences in the trending events that large users (users with many fans and readings) and common users paid attention to during each phase of COVID-19.

## 1. Introduction

Disasters usually happen unexpectedly. Coronavirus disease 2019 (COVID-19), a respiratory disease caused by a novel coronavirus, has spread to over 180 locations internationally in nearly five months since its sudden outbreak in December 2019. According to the World Health Organization (https://www.who.int/emergencies/diseases/novel-coronavirus-2019), as of 22 June 2020, there were 8,860,331 confirmed COVID-19 cases and more than 465,740 deaths worldwide. The effect of COVID-19 on personal health and economic development is self-evident, and it has had a wider impact on public sentiment. The earliest large-scale outbreak of COVID-19 was in China, and the country has now effectively controlled COVID-19. Examining the characteristics of the evolution of public emotion during the COVID-19 outbreak in China would be very useful in helping to understand and predict public emotion worldwide.

To this end, the study aim is to track the evolution of public emotions and analyze the root causes of such emotions from an event-driven perspective. The investigation focuses on the 48 day period following the announcement of the first laboratory-confirmed case of COVID-19 on 1 December 2019 [1] and covers the early and middle stages of the COVID-19 epidemic in China. However, tracking and understanding public emotion evolution during COVID-19 is not an easy task. Three fundamental issues need to be addressed: (1) How can the public’s daily complex emotions be detected and generalized during the disaster? (2) How can the dynamic and constantly changing modes of the public’s complex emotions be captured? (3) What are the criteria for identifying hot events that dominate current public emotions? As COVID-19 evolved, social network sites became appropriate spaces for the social sharing of emotions, opinions, and coping strategies, providing psychological benefits for users by increasing their feelings of emotional relief [2,3,4,5]. Therefore, the study aim is to track public emotion during COVID-19 using the contents of microblogs and to extract the details of dominant events.

There have been several series of investigations of the information flow of social networks during disasters; these have focused on issues such as crowd psychological states [6,7,8], disaster management strategies [9,10], detection of natural disasters [11,12], and public attention to disasters [13,14,15]. Specific to public sentiment analysis, scholars have proposed crowd emotion detection solutions for natural disasters (e.g., earthquakes [16], forest fires, floods, and droughts [17]) based on microblogging contents. If we view the aforementioned research as a pointwise exploration of public emotion at a particular moment caused by short-lived disasters, this study extends the exploration from point to line and adopts a period-wise approach for this longer term disaster situation. The period-wise perspective and long-lasting nature of COVID-19 enable more in-depth research on the dynamic latent connections among complex public emotions, as well as the root causes of public emotions during different phases of the disaster. Embedding public emotion evolution into the skeleton of COVID-19 also made it possible to strip out periodic changes from the overall epidemic process. From this perspective, research on long-term disasters is worthy of attention and is meaningful.

Another difference from previous work is that, despite the special long-term context of COVID-19, this paper considers public emotions to be driven by events. For example, the rapidly rising number of confirmed COVID-19 cases may trigger public panic, whereas effective government control measures may reduce panic. Behaviors that endanger social security may provoke public anger, whereas positive social donations may spark the emotion of pride. Finding the root causes of public emotion evolution would be helpful for understanding and managing the disaster.

The rest of the paper is organized as follows. Section 2 provides a summary of related work. The data collection process and methods of modeling public emotion evolution and extracting hot events are introduced in Section 3. The experimental results are shown in Section 4. The implications and future work are discussed in Section 5.

## 2. Literature Review

### 2.1. Disaster Detection and Management Using Social Networks

Scholars have demonstrated that social media is used extensively for shelter-seeking actions during disasters. Reference [2,3] reported that social media allows users to socially share emotions, seek empathy in a disaster situation, and obtain psychological benefits, as it increases their feelings of emotional relief and being part of a like-minded community. Reference [4,5] reported that data produced through social networks is ubiquitous, rapid, and accessible and has the potential to contribute to situational awareness during emergency social events, not only among civilians, but also among local and federal governments, non-governmental organizations, and local and mass media.

To analyze the effect of a disaster on the psychological state of the public, Reference [7] examined community disaster resilience and social solidarity regarding South Korea’s most tragic maritime disaster, the Sewol ferry disaster, which was repeatedly portrayed on social media. Reference [6] uncovered the dynamics involved in the exchange of social support among a group of Puerto Ricans who experienced a natural disaster and highlighted the importance of culture in shaping patterns of help-seeking behavior in the aftermath of a disaster. For emergency event management, Reference [9] developed a framework for real-time urban disaster damage monitoring and assessment based on social media texts sent during and after the Tianjin explosion and Typhoon Nepartak. Reference [10] explored a mass media crisis information release strategy for controlling public panic stemming from accidental hazardous chemical leakage into rivers. Utilizing social networks as natural disaster sensors, Reference [11] presented and evaluated an earthquake detection procedure that relied solely on Twitter data. Reference [12] analyzed the spatial and temporal characteristics of Twitter feed activity in response to an earthquake in the United States and argued that the feeds represented a hybrid form of a sensor system that allowed the identification and localization of the impact area of the natural disaster. To analyze public attention during disasters, Reference [13] examined how the public discussed measles during a measles outbreak using a semantic network analysis of tweet content. They identified four frames based on word frequencies and co-occurrence: news updates, public health, vaccination, and political. Reference [14] proposed an “attention shift network” to observe, measure, and analyze the collective attention dynamics of Twitter users in response to violent terrorist attacks. In contrast to most previous research, which has concentrated on extracting situational information, Reference [15] focused on the potentially adverse effects of communal tweets during disasters and proposed to filter anti-communal tweets from communal tweets. In addition, Reference [18] presented a learning-based approach to identify prominent microblog users susceptible to sharing relevant and exclusive information in a disaster case. Reference [8] identified microblogging trustworthiness by characterizing the main features of tweets in a crisis, considering the effect of author emotions.

The studies discussed above have focused on natural disasters and social crisis events, including disaster detection, disaster management, public psychological dynamics, and public concerns via social networks. They have mainly focused on sudden, short-lived disasters and subsequent public feedback. In contrast, this study focuses on COVID-19, a disaster that happened suddenly and (to date) has lasted for several months. It is important to identify and assess the trends in hot topics on social networks at different stages of COVID-19, to provide an explanation for the continuous changes in public emotions during the COVID-19 epidemic.

### 2.2. Public Emotion Analysis Using Social Networks

Previous research on sentiment analysis in social networks has generated fruitful results using supervised classifier learning [19,20,21,22] and unsupervised emotion lexicon-based methods [23,24]. Most supervised methods are conducted by extracting linguistic features and constructing classifiers using machine-learning or deep-learning techniques. For example, Reference [19] presented a method to automatically build fine-grained emotion lexicon sets and suggested features that improve the performance of machine-learning-based emotion classification in Korean Twitter texts. However, supervised methods are usually affected by the corpus and limited by complicated parameter settings. In addition, most existing emotion classification methods focus on three polarities (positive, negative, and neutral emotions), the emotional granularity of which is not fine enough [25], and easily experience feature sparseness when extracting textual features from large-scale short and random social network posts [26]. Unsupervised methods detect emotion categories based on the emotional tendency of words in combination with semantics and syntactic structure in the text. For example, Reference [24] introduced an approach to classify emotion in microblog texts as anger, disgust, fear, happiness, liking, sadness, or surprise based on class sequential rules. Reference [23] proposed three pruning strategies to automatically build a word-level emotional dictionary for social emotion detection. In the dictionary, each word is associated with the distribution of a series of human emotions, including anger, disgust, fear, joy, sadness, and surprise. Such lexicon-based unsupervised methods can reduce the amount of calculation, are widely used in fine-grained emotion classification, and largely depend on lexicon coverage.

Specific to the analysis of public emotion during social crises, Reference [27] used an emotional change detection approach to analyze users’ emotions on Twitter in the community involved in the Las Vegas shooting (October 2017). They argued that such analysis could help to improve emergency response services. Reference [28] took a resource perspective in conducting a qualitative analysis of affective expressions on Twitter collected in Germany during the enterohemorrhagic Escherichia coli (EHEC) food contamination incident. Affective expressions of coping went beyond positive or negative tone, as some people perceived the outbreak as a threat, whereas others perceived it as a challenge with which to cope. Reference [9] proposed to measure both urban emotional tendencies and physical damage from social media texts sent during and after the Tianjin explosion and Typhoon Nepartak based on a sentiment dictionary. Reference [29] examined the public mood in Korea in response to the Sewol ferry disaster by analyzing Twitter keywords (“anger”, “anxiety”, and “sadness”). They found that in the aftermath of the incident, keywords related to emotional reactions (such as anger and suicide) were prevalent on Twitter. Reference [30] proposed a disaster assessment model by correlating emotional data in voice and text communications with population density and disaster severity. Reference [31] found that during the 2007 Virginia Tech shooting, anger was the most frequently expressed emotion on Twitter, followed by fright, sadness, and anxiety. In terms of natural disasters, Reference [21] collected microblogs about the “H7N9” influenza and classified them into different emotion categories using a supervised classification method. A topic model was then used to extract subtopics about the event. The authors identified the intensity of five types of emotions for each subtopic over time. Reference [17] collected streaming tweets relating to disasters and built a sentiment classifier to categorize users’ emotions during disasters (earthquakes, forest fires, floods, and drought) based on their varying levels of distress. They subcategorized users’ emotions in response to different disasters as negative sentiments, unhappy, depressed, angry, and positive sentiments. Reference [32] analyzed tweets during Hurricane Sandy and projected users’ sentiments onto a geographical map. Reference [16] aimed to track and predict the changing trends in negative emotions of victims during the post-disaster situation of the Ya’an earthquake. Reference [33] trained a binary sentiment classifier to examine public levels of concern and worry before and after Hurricane Irene. Reference [34] found that one-third of bloggers expressed discernible emotions when blogging about Hurricane Katrina, including concern, disgust, anger, fear, and hope.

Focusing on the process of the evolution of public emotions after crises, Reference [35] constructed a Weibo emotional lexicon with a wide coverage. By analyzing the dependency of Weibo text, they proposed an emotion-computing method based on dependency parsing and computed the emotional tendency and intensity of a single post. They used the Beijing “RYBKindergarten Child Abuse Case” as an example to count the positive and negative emotional posts on Weibo during the three stages of Internet public opinion evolution (outbreak, fermentation, and digestion) and obtained percentages for two types of emotional posts at each stage. Reference [36] qualitatively analyzed online public crisis emotions (especially positive ones) from comments on the Boston Marathon Facebook page during one month following the 2013 Boston Marathon bombing. They conducted qualitative content analysis by reading through comments and coding engagement strategies and emotional tones. They found evidence for positive public emotions; as the crisis evolved, public emotional content, tone, and intensity changed slightly as a result of the online public’s identity and coping and the organization’s engagement. Reference [7] examined how community disaster resilience and social solidarity concerning the Sewol ferry disaster were portrayed by social media over one year. They recorded the most frequently appearing and co-occurring words for three time periods and found a significant change in the sentiments of the community, as well as in ways of forming resilience and solidarity over time. Reference [16] introduced machine-learning methods to track and predict changing emotional trends on microblogs in victims of the Ya’an earthquake. Reference [37] assessed time-specific changes in the intensity and duration of individual distress after an attack, focusing on three primary negative emotions (anger, anxiety, and sadness). Guided by the Crisis and Emergency Risk Communication model, Reference [38] examined public emotional tone (alarm/concern, reassurance, anger, humor/sarcasm, neutral) from 2881 tweets posted during the initial, maintenance, and resolution stages of the 2015 California measles outbreak. They found that the public showed the greatest interest in the initial stage of the crisis, but their interest substantially declined afterward. Reference [39] used the deep-learning method to extract fine-grained public emotions (anger, anxiousness, fear, and sadness) from Chinese microblogs to assist in disaster analysis of the Ya’an earthquake.

These studies are summarized in Table 1. In terms of emotion types, previous work has mainly focused on general public sentiment (positive, negative, or neural) detection (e.g., Reference [9,10,23,25,28]) or negative emotions after crises (e.g., Reference [29,31,33,37,39]). Most research on fine-grained public emotions is restricted to the assessment of emotion types and rarely explores the relationship between different types of public emotions in crises. Regarding the public emotion evolution process, as most crises do not present a continuous evolutionary context, researchers have generally explored the developmental trajectory of public emotions during a specific period of time after the moment of crisis (e.g., Reference [7,16,35,39]). The latent transition modes between different emotions have rarely been studied. Therefore, this article focuses on fine-grained public emotions based on Ekman’s six emotion types. It presents an in-depth exploration of the evolutionary patterns of emotions in different stages of COVID-19, as well as the transitions and co-change relationships among emotions.

## 3. Materials and Methods

### 3.1. Research Overview and Data Collection

In this study, a dataset was constructed using microblog data (n = 125,672) seeded from COVID-19-related super topics (n = 680) from 40,891 users from 1 December 2019 to 17 February 2020, on China’s Sina Weibo platform. All super topics were manually classified into nine categories (“fighting COVID-19”, “novel coronavirus”, “Wenliang Li”, “treatment”, “epidemic development”, “epidemic prevention and control”, “confirmed cases”, “isolation”, and “support Wuhan”) according to their content similarities. Users were divided into two groups: a common user group (n = 38,648) and a large user group (n = 377) based on their account influence after data cleaning. The dataset was then split into subsets by time unit. For each time unit, we aimed to describe the COVID-19 algorithmic skeleton by extracting and summarizing microblogging content in two dimensions: the number of newly confirmed COVID-19 cases and the number of accumulated COVID-19 cases. The hot events in each subset were then hierarchically extracted based on criteria related to both the microblogging and super topic characteristics. To automatically extract public emotions, we built upon previous emotion detection work and modeled the complex emotions in each time unit as a multidimensional description. We followed the categorization of the Chinese Affective Vocabulary Ontology Database for seven dimensions (joy, anger, sadness, fear, disgust, love, surprise) and 21 subdimensions, covering 27,466 Chinese phrases [41]. Weighted term frequency-inverse document frequency (tf-idf) measures were applied to construct the complex emotion matrix for each time unit. Based on emotion matrices, the accumulated emotion trends, the latent covariance modes, and the continuous transition modes for each pair of public emotions were modeled during different phases of COVID-19.

*Microblogs*. As China’s largest and the only active public social network platform, Sina Weibo fosters user relationships that allow users to share, disseminate, and receive information. Through either the website or the mobile app (https://en.wikipedia.org/wiki/Sina_Weibo), users can publish microblogs containing text or multimedia contents for instant sharing; other users are able to browse, comment, or forward the content (similar to “retweet” in Twitter). Sina Weibo attracts more than 560 million monthly active users. Its user group shows a trend for younger users; 25% of users are of the post-1980s generation, 29% of the post-1990s generation, and 37% of the post-1995 generation (https://tui.weibo.com/insight/detail?id=222). In this study, the microblog dataset was collected from two sources. First, a search was made for keywords (n = 12) related to COVID-19 on the Sina microblog platform, and super topics containing the keywords were extracted. The microblogs (n = 93,840) labeled by these super topics were collected and posted by 37,108 users. This source ensured a large range of COVID-19-related topics (n = 585). Second, to ensure large coverage of important news during COVID-19, we examined the above dataset, filtered out the four most popular official accounts (People’s Daily, Global Times, People’s Hotspots, CCTV News), and collected their microblogs (n = 2183) from 1 December 2019 (when the first laboratory-confirmed case was announced in Wuhan [1]) to 17 February 2020. These four accounts have continued posting COVID-19-related news every day since the COVID-19 outbreak and contain 95 super topics. According to these 95 super topics, thirty-two thousand two-hundred seventeen microblogs were collected and combined into the dataset. We filtered out super topics with fewer than 50 microblogs and limited the post time to between 1 December 2019 and 17 February 2020. Finally, the dataset contained 125,672 microblogs, which were labeled with 326 super topics from 40,891 users. The data collection process is shown in Table 2. All emojis or emoticons in microblog content were converted into corresponding text (e.g., [Smile]).

*Super topics*. All super topics were manually divided into nine categories based on their content, as shown in Table 3 and Figure 1. The number of super topics and microblogs, as well as the duration of each category are also shown. For example, under the category “confirmed cases”, a total of 166 super topics and 68,555 microblogs were collected (e.g., #A total of 14,380 confirmed cases nationwide#). These nine super topic categories reflected news covering (1) measures that people adopted to fight COVID-19, (2) the process of COVID-19 in China, (3) a hero who appeared during the disaster, (4) treatment knowledge and achievements, (5) epidemic development nationwide, (6) the number of confirmed cases, (7) support from the world to Wuhan, (8) isolation, and (9) official epidemic control programs.

*Users*. Users were divided into two groups: a large user group and a common user group. We marked users who had ever posted microblogs with more than 1000 comments in the collected dataset as large users. We used this classification because the microblogs of the two groups reflected different information. Common users tended to express their thoughts on the current super topic, and the number of comments, retweets, and likes for their microblogs were relatively small. Large users tended to post important official news, which attracted tens of thousands of fans and substantial social attention, but the content of these posts lacked self-expressed information. The mean number of retweets, comments, likes, topic discussions, and topic reads in each group are shown in Table 4.

### 3.2. Models

#### 3.2.1. Extracting the Skeleton of COVID-19

The skeleton describing the evolution process of COVID-19 was defined in two dimensions. Let $n_{i}^{new}$ be the number of newest confirmed cases in time unit $t_{i}$ (day) and $n_{i}^{acc}$ be the number of accumulated confirmed cases until time unit $t_{i}$. $<n_{i}^{new},n_{i}^{acc}>$ describes the COVID-19 cases in time unit $t_{i}$. Thus, $<n_{1}^{new},n_{1}^{acc}>,<n_{2}^{new},n_{2}^{acc}>,\cdots ,<n_{m}^{new},n_{m}^{acc}>$ describes the skeleton of COVID-19 in period $<t_{1},t_{2},\cdots ,t_{m}>$. Regular expressions in Python were utilized to match $n_{i}$ and $N_{i}$ from microblogging contents of each time unit in the dataset. For a time unit for which multiple $<n_{i}^{new},n_{i}^{acc}>$ results were extracted, we ranked all results and chose the largest number for each dimension, considering that the confirmed number may vary and be reported more than once during a time unit. The final results were manually confirmed to ensure no conflicts and are represented in Figure 2. The extraction results before 21 January 2020 are not shown owing to the low posting rates. The number of new confirmed cases continued to increase from 21 January 2020 to 2 February 2020 and kept increasing until 12 February 2020. The peak appeared on February 12 and then kept decreasing until 17 February 2020. Thus, we divided the COVID-19 evolution into three phases in the following analysis according to these two change points.

#### 3.2.2. Evolution Models of Complex Public Emotions

To extract complex public emotions during COVID-19, the microblog dataset was divided into *k* subsets based on *k* time units and clustered into *k* documents, denoted as $D=<d_{1},\cdots ,d_{k}>$. For a document *d*, we aimed to identify its emotion distributions and sub-emotion distributions. The complex emotions in each time unit were extracted according to the Chinese Affective Vocabulary Ontology Database [41]. This vocabulary follows the categorization of Ekman’s universal emotions [42] on six dimensions C = {joy, anger, sadness, fear, disgust, surprise} and, in particular, explores the emotion love based on a Chinese corpus containing 27,466 Chinese phrases on 21 subdimensions. Each emotional word *w* was labeled by <category, subcategory, degree>. For clarification, the following is an example of the extraction of emotion distributions; the extraction of sub-emotions followed a similar process.

To identify the most obvious emotions of each time unit, we propose a weighting scheme for the emotional words in each document $d_{i}\in D$ based on the tf-idf method. The tf-idf value increases proportionally to the number of times an emotional term $w^{emo}$ appears in the document *d* and is offset by the number of documents in the corpus *D* that contain the term $w^{emo}$, thus adjusting for the fact that some words generally appear more frequently. The frequency of the emotional term $w_{i}^{emo}$ in document $d_{j}$ is denoted as follows:(1)$$tf_{ij}=\frac{n_{i,j}}{\sum_{\forall w_{p}^{emo}\in d_{j}}n_{p,j}}$$
where $n_{i,j}$ is the normalized frequency of term $w_{i}^{emo}$ in document $d_{j}$. The inverse document frequency of emotional term $w_{i}^{emo}$ and the tf-idf value of $w_{i}^{emo}$ in document $d_{j}$ are denoted as follows:(2)$$\begin{array}{cc} idf_{i}=log\frac{\left|D\right|}{|\{j|t_{i}\in d_{j}\wedge d_{j}\in D\}|+1} & \\ tfIdf\left(w_{i}^{emo},d_{j}\right)=tf_{ij}\times idf_{i} \end{array}$$

Furthermore, the emotional degree of each word was combined with the tf-idf value to weight its emotional importance for the current document. Each emotional word $w_{i}^{emo}$ in the vocabulary was assigned a degree level $L\left(w_{i}^{emo}\right)\in \left\{1,3,5,7,9\right\}$. Let *L* × $tfIdf(w_{i}^{emo},d_{j})$ be the emotional weight of $w_{i}^{emo}$ contributed to document $d_{j}$. For each category c ∈C, its emotional importance in document $d_{j}$ is denoted as follows:(3)$$Ratio\left(c\right)=\frac{\sum_{\forall w_{i}^{emo}\in d_{j}\wedge c_{i}=c}L\left(w_{i}^{emo}\right)\times tfIdf\left(w_{i}^{emo},d_{j}\right)}{\sum_{\forall w_{p}^{emo}\in d_{j}}L\left(w_{p}^{emo}\right)\times ifIdf\left(w_{p}^{emo},d_{j}\right)}$$
Thus, the complex emotions during the *k* day period of COVID-19 are represented as a *k* × 7 matrix M, and the complex sub-emotions are represented as a k×21 matrix $M_{sub}$.

**Accumulated Emotions:** The accumulated value represents the change in the intensity of each category of public emotion during COVID-19. To eliminate the influence of the unbalanced amount of microblogs in each time unit, $tf_{ij}$ is adopted as the frequency of the emotional term $w_{i}^{emo}$ in document $d_{j}$, where $d_{j}$ is the document aggregated from microblogs posted within the time unit $t_{j}$ (day). For each emotion category c∈C, its accumulated value in each time unit $t_{j}$ is denoted as:(4)$$Acc\left(c\right)=\sum_{\forall w_{i}^{emo}\in d_{j}\wedge c_{i}=c}L\left(w_{i}^{emo}\right)\times tf_{ij}$$
where $L\left(w_{i}^{emo}\right)\in \left\{1,3,5,7,9\right\}$ is the degree level of emotional word $w_{i}^{emo}$ assigned in the vocabulary.

**Emotion Covariance Model:** The covariance model measures whether two public emotion categories changed according to the same or opposite trends during COVID-19. To find the latent covariance modes between each pair of emotions, based on the complex emotion matrix M and the sub-emotion matrix $M_{sub}$, we extracted the emotion covariance accompanying the COVID-19 skeleton (see Section 3.2.1). In Figure 2, the COVID-19 period from 21 January 2020 to 17 February 2020 was divided into three phases by the two change points of new confirmed cases: 21 January 2020 to 4 February 2020; 5 February 2020 to 12 February 2020; and 13 February 2020 to 17 February 2020, denoted as *I* = $<I_{1},I_{2},I_{3}>$. For clarification, the following takes the emotion matrix M and phase $I_{1}$ as an example. The covariance between each pair of sub-emotions in $M_{sub}$ during the other subperiods $I_{2}$ and $I_{3}$ follows the same process. For each two emotion categories $c_{i}$, $c_{j}$∈C, and vectors $C_{i}$ and $C_{j}$ are extracted from emotion matrix M, $|C_{i}\left|=\right|I_{1}|$, $|C_{j}\left|=\right|I_{1}|$. The Pearson correlation coefficient between $C_{i}$ and $C_{j}$ is used to represent the covariance between emotion $c_{i}$ and $c_{j}$ in the current phase during COVID-19.

**Emotion Transition Model:** An emotion transition model extended from previous work [43] was used to measure whether there was mutual transformation between the two public emotion categories. We used the emotion matrix M during phase $I_{1}$ as an example. First, all the emotion vectors $C_{i}$ in M are first converted to ranked vectors by category, denoted as $C_{i}^{r}$ and $M^{r}$, respectively. Thus, the effects of emotional imbalances are eliminated. Next, the emotional transitions between two neighboring time units $U_{i}$ and $U_{i+1}$ are measured. *U* is a 7×1 vector here. For each category of emotion $U_{i,p}$ in the ranked emotional vector of time unit $U_{i}$, we compared $U_{i,p}$ with each emotion category in the next time unit $U_{i+1}$, applying function comp(x,y) as follows:(5)$$comp\left(x,y\right)=\left\{\begin{array}{c} 1,\;ifx=y,\\ 0,\;otherwise \end{array}\right.$$
Thus, for each pair of neighboring time units $U_{i}$ and $U_{i+1}$, the transition matrix $T_{i,i+1}$ = $Trans(U_{i},U_{i+1})$ is denoted as follows:(6)$$T_{i,i+1}\left(p,q\right)=comp\left(U_{i,p},U_{i+1,q}\right)$$
where $i\in [1,|I_{1}|-1]$, p∈[1,7], q∈[1,7]. The public emotion transition matrix in phase $I_{1}$ during COVID-19 is denoted as follows:(7)$$\begin{array}{cc} T & =Trans(U_{1},\cdots ,U_{|I_{1}|-1})\\ \; & =\sum_{i\in [1,|I_{1}|-1]}Trans\left(U_{i},U_{i+1}\right) \end{array}$$

#### 3.2.3. Extracting Hot Events from COVID-19 Super Topics

Hot events are hierarchically extracted according to their popularity in each time unit. For each super topic *e*, let *W*$=<w_{1},\cdots ,w_{n}>$ be the set of microblogs labeled by super topic *e*. Five measures are used to describe the hot degree hot(e) from two aspects. (1) At the microblog level, for each post $w_{i}$, the number of retweets, the number of comments, and the number of likes are used to describe the hot degree of *e*, denoted as $H_{i}=\left[w_{i}^{retweet},w_{i}^{comment},w_{i}^{like}\right]$. (2) At the topic level, the number of discussions and the number of reads are used to describe the hot degree of *e*, denoted as $S^{e}=\left[e^{discuss},e^{read}\right]$. For the set of super topics $E=<e_{1},\cdots ,e_{k}>$ in each time unit, the hot degree of each super topic *e* is as follows:(8)$$\begin{array}{c} hot\left(e\right)=weight_{1}*||rank\left(\frac{1}{n}\sum_{i\in [1,n]}H_{i}\right){\left|\right|}_{1}+weight_{2}*||rank\left(S^{e}\right){\left|\right|}_{1} \end{array}$$
Thus, the hottest event in the current time unit is denoted as follows:(9)$$\begin{array}{cc} \; & H\left(E\right)=argmin_{e_{i}\in E}hot\left(e_{i}\right) \end{array}$$

This paper represents the results of the top two super topics for each time unit in Section 4 and sets $weight_{1}=weight_{2}=0.5$.

## 4. Results

To better explain the experimental results, we first introduce the detection results for hot events during COVID-19 in Section 4.1. Then, we show the detection results for public emotion evolution and explain public emotion changes in Section 4.2 based on the corresponding hot social events.

### 4.1. Hot Events from Super Topics during COVID-19

Hot events among common users and large users were extracted by time unit. The top three hot events were statistically analyzed and are shown in Figure 3. From 1 January 2020 to 17 February 2020, among common users, twenty-six-point-eight-three percent of the hot events concentrated on topics about the “New coronavirus”, followed by “Isolation” at 20.33% and “Epidemic development” at 17.07%. For large users, only one super topic “Wuhan unknown cause pneumonia” was found before 21 January 2020; thus, the hot events were analyzed from 21 January 2020. These hot events concentrated on topics about “Confirmed cases”, which achieved a high proportion of 48.19%, followed by “Supporting Wuhan” at 13.25%. The top three hot events at each time unit at different phases of COVID-19 for common users and large users are shown in Figure 4a. The area of each circle represents the popularity of the current event (e.g., the top hot event is denoted as the largest circle), and the nine event categories are differentiated by color. As few super topics were identified at the very start of the COVID-19 period, before 21 January 2020, we viewed this period of time as a warm-up phase and did not include this period in the subsequent analysis. The detection results for category “Wenliang Li” are not shown here owing to their sparsity. The Y axis shows the number of discussions logged under each hot event. To highlight this pattern, only the top three events are shown. Table A1 and Table A2 (see Appendix A) list the top and top two hot events in nine groups at each time unit for common users and large users from 21 January 2020 to 17 February 2020.

### 4.2. Public Emotion Evolution

In this section, we show the detection results for public emotion evolution in different phases of COVID-19 from three aspects: accumulated emotions, emotion covariances, and emotion transitions. We explain public emotion changes based on the corresponding hot social events during different phases of COVID-19 in Section 4.1.

**Accumulated Emotions During COVID-19**. The accumulated values of seven public emotions and 21 sub-emotions during COVID-19 were analyzed and are presented in Figure 5. Love appeared to have the most significant accumulated value during the whole period from 21 January 2020 to 17 February 2020. The accumulated value of love increased around several time points, such as 26 January 2020, 9 February 2020, and 14 February 2020. It is worth noting that from 14 February 2020 to 16 February 2020, the accumulated value of love was very high. The accumulated value of sub-emotion PD(respect) increased several times, especially from 14 February 2020 to 16 February 2020. PH(praise) showed relatively high accumulated values during the observation window.

**Emotion Covariances in Different Phases of COVID-19**. The covariances among complex public emotions during COVID-19 were analyzed and are presented in Figure 6 by phase. Each grid of coordinates (x, y) represents the covariance coefficient of two categories of emotions x and y for seven emotions. To avoid confusion, public emotion covariance was analyzed by three phases according to peak/valley values on the “newest confirmed cases of COVID-19” skeleton from 21 January 2020 to 17 February 2020. In the first phase (21 January 2020 to 4 February 2020), the sadness and disgust emotions appeared to have the most significant covariance. In the second phase (4 February 2020 to 12 February 2020), although the covariance between sadness and disgust persisted, the new main covariance was between surprise and joy. Surprise was highly covariant with joy and anger. In the third phase (12 February 2020 to 17 February 2020), stronger covariance between positive and negative emotions occurred. In contrast with the previous two phases, a strong negative correlation appeared between sadness and surprise in the third phase. The covariances for the 21 sub-emotions are presented in Figure A1. The sub-emotions of (doubt, reprimand) and (reprimand, wishing) showed significant covariance during the whole COVID-19 period.

**Emotion Transitions in Different Phases of COVID-19**. The transitions between complex public emotions are presented in Figure 6 and analyzed by phase from 21 January 2020 to 17 February 2020. The number in each grid (x, y) denotes the frequency of the accumulated transitions between emotion x and emotion y at two neighboring time units. Across the three phases, love was the most frequent and stable emotion. Disgust was the most stable emotion during the first phase, and anger was the most stable emotion during the second phase. Transitions between (anger → surprise) appeared frequently during the first phase (21 January 2020 to 4 February 2020). Transitions between (sadness → disgust) were relatively high during the third phase. The transitions between 21 sub-emotions are presented in Figure A1. The sub-emotions (jealousy → reassurance) frequently exhibited transitions, and the sub-emotion praise showed high continuity during COVID-19.

## 5. Discussion

The aim of this research was to use social networks to track the public’s emotions during the spread of COVID-19 over a 48 day period since the first laboratory-confirmed case was announced on 1 December 2019, in China, and to analyze hot spot events underlying the emotions. The results show that, at different stages of COVID-19, public emotions showed interesting migration trajectories, as evidenced by both the covariance and transition modes. There was also a clear difference in the focus and reactions of common microblog users and official accounts in the face of the epidemic.

Some studies on public reactions to crises have focused on negative emotions (e.g., anger, anxiousness, fear, and sadness [37,39]) and have considered the threat of negative emotions to institutional trust [44], reputation [45], and the contribution to responsibility [45]. However, our findings show that public positive emotions (e.g., love, respect, praise) co-existed with and even surpassed negative emotions during COVID-19, which is consistent with previous research on other crisis situations [45,46,47]. Love appeared to have the most significant accumulated value during the whole period of 21 January 2020 to 17 February 2020 and increased around several time points (26 January 2020; 9 February 2020; and 14 February 2020). We further checked top-ranked hot events (Table A1 and Table A2) and found that social events about “Fighting COVID-19” became hot during this period, as well as good news about effective treatment (e.g., “Cumulative total of 5911 cases discharged from hospital”) and concerns about new cases (e.g., “Total number of cases successfully admitted to Vulcan Mountain Hospital exceeds 1000”). The sub-emotions Respect (PD) and Praise (PH) also showed high accumulated values and increased several times, especially from 14 February 2020 to 16 February 2020. The results confirmed that prevalent positive emotions can enhance the public’s coping function during disasters [48,49] and increase trust in government [36,50].

According to [51,52], public reactions to crises comprise four main coping methods: rational thinking (cognitive), emotional venting (emotional), instrumental support, and action (conative). At the very beginning phase of COVID-19 (before 21 January 2020), while instrumental support and effective actions from the government were lacking owing to the suddenness of the epidemic, the public’s feedback on COVID-19 predated official accounts on the microblog platform and focused on “isolation” and “novel coronavirus”. The microblog platform was used to share timely information and to create a sense of connectedness and functioned as a “psychological first aid” for the public [53]. Then, in the first phase (21 January 2020 to 4 February 2020), the emotions of (sadness, disgust) showed the most significant covariance. The sub-emotions (doubt, reprimand) and (reprimand, wishing) also showed significant covariance at this time. The public mainly focused on “Isolation” and “Rapid development of epidemic”. The public’s rational thinking and emotional venting were also captured, as topics about “Support Wuhan” became hot among both common and large users. During the middle phase (4 February 2020 to 12 February 2020), surprise and joy showed a new main covariance. The emergence of this covariance can be attributed to several causes. For example, with the decrease in daily reported new confirmed cases, topics about effective “Treatment” and “Fighting COVID-19” (e.g., “Original relay to fight COVID-19”) became hot in this phase (Figure 4). Topics about “Epidemic prevention and control” became hot among large users. Topics about instrumental support (providing information and social support [51]) from governments, non-governmental organizations, and medical staff across the country were extremely hot, particularly helped to boost public crisis coping and positive emotions, and also reflected the community disaster resilience of the government [54].

It is notable that surprise was also covariant with anger during the middle phase (4 February 2020 to 12 February 2020). This may have been triggered by the fact that top-ranked events about “Doctor Wenliang Li” appeared among common users (see Table A1 and Table A2). Doctor Wenliang Li was the first person to issue a protective warning about COVID-19 and was called the “whistleblower” of the epidemic, but was unfortunately infected. In the last phase (12 February 2020 to 17 February 2020), a strong negative correlation appeared between sadness and surprise. Discussions about “Decrease in newly confirmed cases” became hot in this phase. In addition, after checking all event ranking results in this phase, we found that more than 20 super topics were proposed about the death of Doctor Wenliang Li. Research shows that community members tend to pay attention to victims and their symbolic behavior after a tragedy, and such social solidarity can increase the collective response to disasters [55]. Communal bereavement can also increase social solidarity and lead to collective action, such as memorial rituals [56]. This could explain why the public expressed such strong sadness for Wenliang Li, as well as anger and disgust at his tragic death.

Considering the emotion stability, love persisted as the most stable emotion during the whole COVID-19 period. Disgust was the most stable emotion during the initial phase, and anger was the most stable emotion during the second phase. Considering the emotion transition patterns, transitions between the emotions (anger → surprise) and the sub-emotions (jealousy→ reassurance) frequently occurred. Throughout the observation window, the topics “Novel coronavirus” and “Confirmed cases” remained hot, but the number of discussions about these two categories gradually reduced over time, reflecting the process of gradual public adaptation to sudden disasters. This reflects one of the psychological functions of social media usage in disasters, which is to increase the public’s feeling of emotional relief [2,3,4,5]. This finding also indicates that disaster resilience [57,58] and social solidarity [59,60] during COVID-19 in China occurred at different levels (e.g., individual, institutional, and environmental) from the perspective of continuous semantic analysis.

Another issue that requires consideration is the role of government censorship of the Sina Weibo platform. According to the Sina Weibo Service Use Agreement (https://www.weibo.com/signup/v5/protocol), in cooperation with Internet censorship in China, Sina Weibo imposes some restrictions on the content posted by users; for example, “Must not violate the laws and regulations of the People’s Republic of China and relevant international treaties or rules” (Regulation 4.10.1) and “Do not upload, display or disseminate any false, racially discriminatory, violent, bloody, or other illegal information materials.” (Regulation 4.10.4). In addition, news media and government agencies that open Weibo accounts must also abide by relevant laws, regulations, organizational rules, and regulatory requirements (Regulation 4.2). Such censorship ensures a safe and healthy operating environment for the platform and does not normally restrict users from publishing legal content. Our dataset and experimental results also show that during COVID-19, negative public emotions, such as doubt, jealously, and disgust, were expressed on the platform. Therefore, Sina Weibo can be regarded as a social platform that can objectively reflect public emotions during crises.

This was an exploratory study and therefore had some limitations. First, we did not differentiate between sarcasm and true negative emotions in the microblogging contents. Sarcasm is a special kind of sentiment that uses expressions that mean the opposite of what the person really wants to say [61]. In recent years, the automatic detection of sarcasm in social networks has been considered an interesting research topic in the field of information retrieval and natural language processing. However, it is also recognized as a difficult task [62], as it requires a system that has the necessary knowledge to interpret the linguistic styles of authors [63]. Researchers have mainly used machine-learning methods [64,65,66] and lexicon-based methods for sarcasm detection. For example, Reference [63] attempted to detect sarcasm in microblogs by considering different sets of features (function words and parts of speech n-grams) and tested a range of different feature sets using fuzzy clustering and naive Bayesian models. Reference [61] proposed two approaches to detect sarcasm in the text of Twitter data: a parsing-based lexicon generation algorithm and the occurrence of interjection words. Reference [62] used a hybrid approach by combining machine-learning and lexicon-based methods. Specific to our problem, the addition of sarcasm detection in future studies could help to improve the accuracy of public emotion detection.

Second, regarding the Sina Weibo platform used in this study, an important issue is how to differentiate between real users who retweeted the microblogs and commented, liked, or read the sites from paid bots. At present, both the platform and researchers have proposed a series of solutions to recognize bots. The Sina Weibo platform has implemented complex technical rules to determine whether user account behavior constitutes an automated behavior; for example, frequent repeated postings, a large number of zombie fans, forwarding, but no original microblogs, no interaction with others, and never commenting. Such accounts are assumed to be “bots” (https://new.qq.com/omn/20181113/20181113A0TTA1.html?pc, https://www.weibo.com/signup/v5/protocol). In addition, researchers have conducted much in-depth research on the detection of bots in microblogs and tweets, applying multi-feature-based recognition methods and spam-filtering techniques [67,68,69,70]. The present study did not include the identification of bot accounts as a research aim. The elimination of such fake accounts in future work would help to improve data quality and to obtain more accurate research results.

Third, this study explored general public emotion evolution during COVID-19 in China. In practice, the situation may differ considerably in different regions of the country in terms of, for example, prevention and control policies, time of the outbreak, and public feedback, especially for severely affected cities like Wuhan. Owing to differences in the effect of the epidemic, the emotional states of Wuhan citizens may be different from those in other regions of the country. Therefore, more detailed information (e.g., positioning information published with microblogs, regional information in profiles) needs to be collected to carry out in-depth research on public emotion evolution in different regions. COVID-19 was continuing to evolve when this paper was last modified in August 2020. More areas of the world are facing serious situations because of COVID-19, and the psychological states of the public around the world are attracting more attention. Future work also needs to consider various factors like culture differences and government management.

## 6. Conclusions

This study focused on the dynamic latent connections among complex public emotions during COVID-19 in China and viewed public emotions as driven by events. To strip out periodic changes from the overall epidemic process, public emotion evolution was embedded into the skeleton of COVID-19 and explained according to hot spot events. This process can help to identify the root causes of public emotions during different phases of the disaster. We collected microblogs during a 48 day period from when the first laboratory-confirmed case was announced on 1 December 2019, in China. Several key turning points in the occurrence, development (the number of confirmed cases continued to rise), and effective control (the number of confirmed cases continued to decline) of COVID-19 were identified. As COVID-19 is still developing, this study could not analyze the follow-up situation. However, the analysis of these key turning points has important reference value for the understanding of the epidemic situation in other countries experiencing COVID-19.

## Figures and Tables

**Figure 1 ijerph-17-06888-f001:**
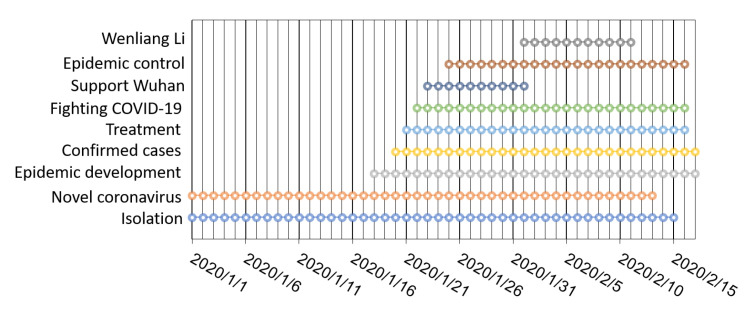
Nine super topic groups and corresponding intervals related to COVID-19.

**Figure 2 ijerph-17-06888-f002:**
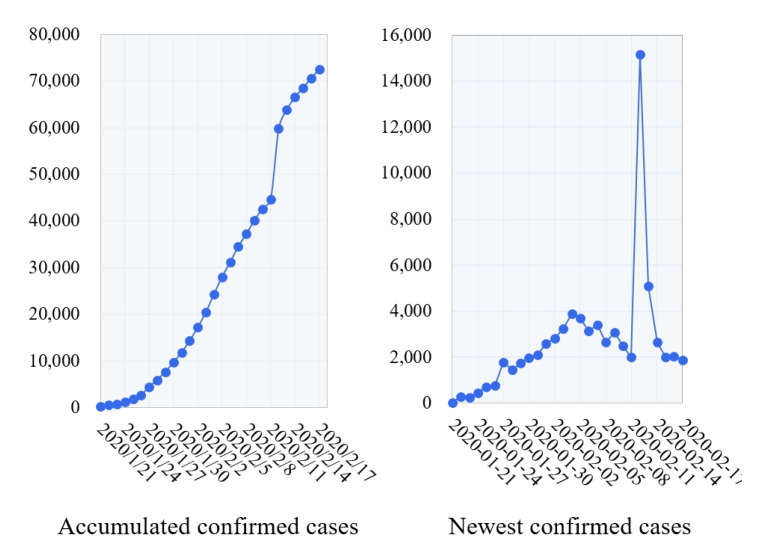
Confirmed COVID-19 cases extracted from microblogs from 21 January 2020 to 17 February 2020.

**Figure 3 ijerph-17-06888-f003:**
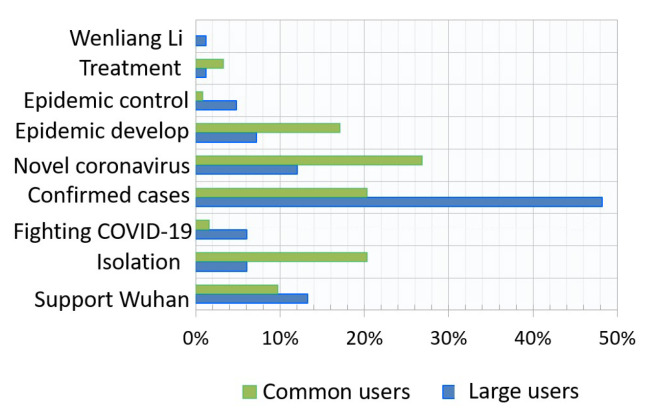
Hot event (topics people were talking about) distributions among common users and large users during COVID-19.

**Figure 4 ijerph-17-06888-f004:**
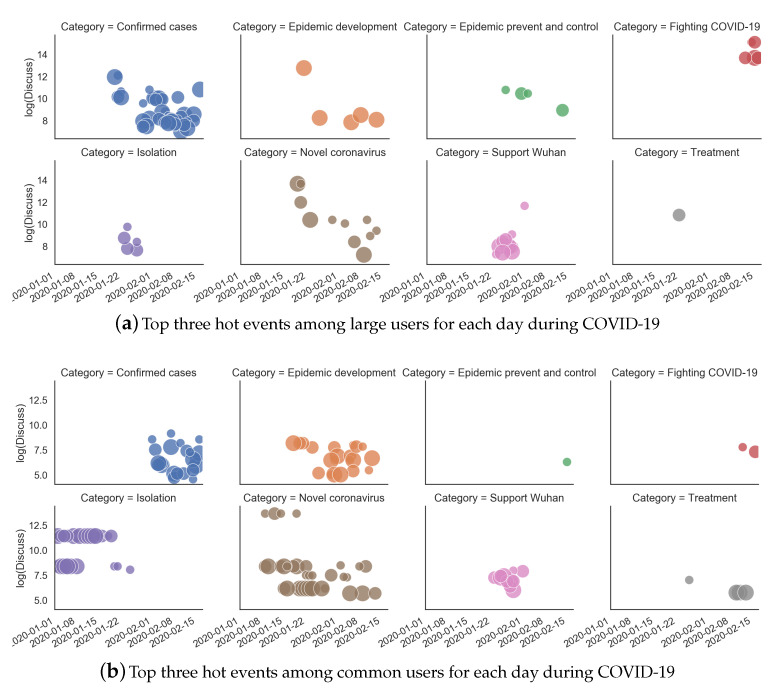
Hot events during COVID-19 extracted from microblog super topics from 1 January 2020 to 17 February 2020.

**Figure 5 ijerph-17-06888-f005:**
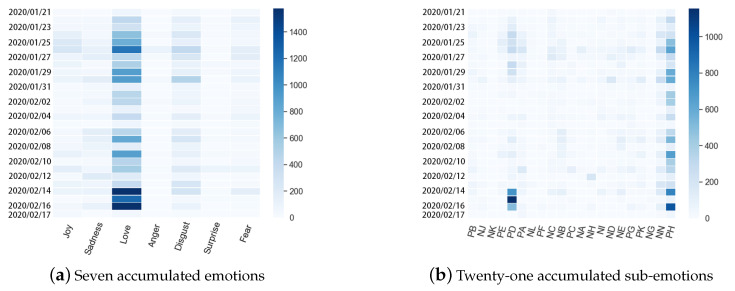
Public accumulated emotions from 21 January 2020 to 17 February 2020.

**Figure 6 ijerph-17-06888-f006:**
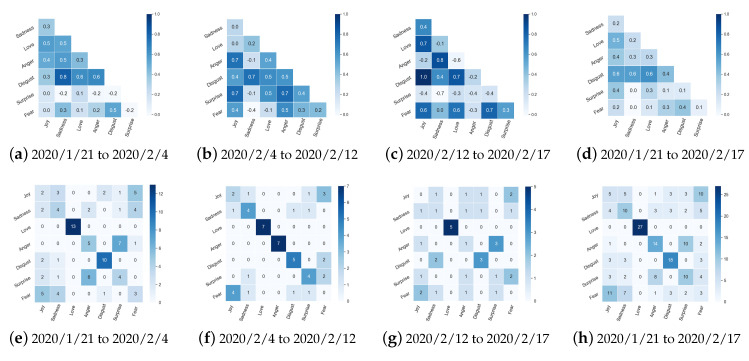
Public emotion covariance and transitions from 21 January 2020 to 17 February 2020, divided into three phases according to the peak/valley values of “newest confirmed cases of COVID-19”.

**Table 1 ijerph-17-06888-t001:** Literature related to emotion analysis in social network communication.

Platform	Emotion Types	Public Emotion	Crisis Related	Emotion Evolution	Method	Literature
Korean Twitter	Ekman’s six emotion types	No	No	No	Machine-learning	[19]
Twitter	Plutchik’s “wheel of emotions”	No	No	No	Machine-learning	[22]
Chinese microblog	Positive, negative	No	No	No	Machine-learning	[20]
Chinese microblog	Anger, disgust, fear, happiness, like, sadness, or surprise	No	No	No	Rule based	[24]
Chinese and English news corpus	Ekman’s six emotion types	No	No	No	Rule based	[23]
Facebook	Ekman’s six emotion types	Yes	No	No	Machine-learning	[40]
Chinese reviews corpus	Positive, negative	Yes	No	No	Machine-learning	[25]
Chinese and English reviews corpus	Positive, negative, neural	Yes	No	No	Deep-learning	[26]
Twitter	Plutchik’s “wheel of emotions”	Yes	Las Vegas shooting	No	Lexicon based	[27]
Twitter	Positive, negative, threat	Yes	EHEC incident	No	Manual coding	[28]
Chinese news corpus	Positive, negative, threat	Yes	Hazardous chemical leakage	No	Rule based	[10]
Chinese microblog	Positive, negative	Yes	Tianjin explosion, Typhoon Nepartak	No	Lexicon based	[9]
Twitter	Negative (unhappy, depressed, angry) and positive	Yes	Yes	No	Machine-learning	[17]
Twitter	Worry, irrelevant	Yes	Hurricane Irene	No	Machine-learning	[33]
Twitter	Anger, anxiety, sadness	Yes	Sewol ferry disaster	No	Inductive methods	[29]
Messages	Sadness, joy, anger, fear, and desperation	Yes	Yes	No	Rule based	[30]
Twitter	Anger, fright, anxiety, and sadness	Yes	2007 Virginia Tech shooting	No	Manual coding	[31]
Blogs, news	Concern, disgust, anger, fear, and hope	Yes	Hurricane Katrina	No	Manual coding	[34]
Twitter	Ekman’s six emotions, neutral, irrelevant	Yes	Hurricane Sandy, Boston bombings	No	Online survey	[8]
Facebook	Positive	Yes	Boston Marathon bombing	Yes	Inductive methods	[36]
Chinese microblog	Happiness, anger, fear, sadness, and surprise	Yes	H7N9	Yes	Machine-learning	[21]
Chinese microblog	happy, sad, fearful, disgusted	Yes	RYBKindergarten Child Abuse	Yes	Deep-learning	[35]
Chinese microblog	Positive, negative	Yes	Ya’an earthquake	Yes	Machine-learning	[16]
Korean Naver blog	Sympathy, positive, hope	Yes	Sewol ferry disaster	Yes	Inductive methods	[7]
Twitter	Anger, anxiety, and sadness	Yes	Yes	Yes	Rule based	[37]
Chinese microblog	Anger, anxiousness, fear, and sadness	Yes	Ya’an earthquake	Yes	Deep-learning	[39]
Twitter	Alarm/concern, reassurance, anger, humor/sarcasm, neutral	Yes	2015 California measles	Yes	Inductive methods	[38]

1. Ekman’s six emotion types comprise “happiness”, “sadness”, “anger”, “disgust”, “surprise”, and “fear”. 2. Plutchik’s “wheel of emotions” comprises “anger”, “disgust”, “fear”, “happiness”, “sadness”, “surprise”, “trust”, and “anticipation”. 3. ECEC incident is the abbreviation for the enterohemorrhagic Escherichia coli food contamination incident in 2011. 4. H7N9 is the abbreviation for Influenza A virus subtype H7N9 (a bird flu strain of the species Influenza virus A).

**Table 2 ijerph-17-06888-t002:** Characteristics of the COVID-19-related microblog datasets and data cleaning processes.

Data and Cleaning Process	No. of Super Topics	No. of Users	No. of Microblogs	Post Date
$Dataset_{keywords}$	585	37,108	93,840	-
$Dataset_{official}$	95	3,783	32,217	1 December 2019–17 February
$Dataset_{keywords}\cup Dataset_{official}$ filtered by time	678	40,710	126,055	1 December 2019–17 February
$Dataset_{keywords}\cup Dataset_{official}$ filtered by topic	326	39,025	125,672	1 December 2019–17 February

1. $Dataset_{keywords}$ was collected based on 12 keywords (new pneumonia, confirmed diagnoses nationwide, epidemic development, novel coronavirus, epidemic map, isolation, support Wuhan, Wenliang Li, coronavirus, South China Seafood Market, new cases nationwide, and new pneumonia cases). 2. $Dataset_{official}$ was collected according to super topics posted by four popular official accounts (People’s Daily, Global Times, People’s Hotspots, and CCTV News).

**Table 3 ijerph-17-06888-t003:** Characteristics of nine super topic categories and microblogs.

Category	No. of Super Topics	No. of Microblogs	Duration (Days)	Examples of Super Topics
1. Fighting COVID-19	6	6431	25	#Fight the coronavirus together#
				#Fighting new pneumonia we are in action#
2. Novel coronavirus	31	11,231	49	#6 key issues of new coronavirus#
				#Source of new coronavirus is wildlife#
3. Wenliang Li	28	5477	10	#Who remembers Li Wenliang#
				#Li Wenliang dies of new pneumonia#
4. Treatment	10	2784	26	#34 cases of new pneumonia have been cured#
				#Tribute to front line health care workers#
5. Epidemic development	33	12,048	30	#Latest national epidemic map#
				#The epidemic is still spreading#
6. Confirmed cases	166	68,555	28	#A total of 14,380 confirmed cases nationwide#
				#1280 new cases of new pneumonia nationwide#
7. Support Wuhan	30	5197	9	#Japan donates 1 million masks to help Wuhan#
				#A medical team of 1230 people assists Wuhan#
8. Isolation	16	3084	76	#333 passengers in Thailand from Nanjing were isolated#
9. Epidemic control	8	3663	22	#Beijing Epidemic Prevention and Control Conference#

1. Wenliang Li was a doctor who made an early effort to call the public’s attention to COVID-19. He died of COVID-19 on 7 February 2020.

**Table 4 ijerph-17-06888-t004:** Characteristics of nine microblog categories posted by large users and common users.

Large users	*C* _1_	*C* _2_	*C* _3_	*C* _4_	*C* _5_	*C* _6_	*C* _7_	*C* _8_	*C* _9_
Mean No. of retweets	27,375	7361	26,236	5824	6289	3119	6000	6964	7438
Mean No. of comments	11,479	5301	28,308	7041	5558	7350	8840	5779	5818
Mean No. of likes	76,789	136,685	298,206	192,633	137,032	143,591	178,286	123,502	172,849
Mean No. of topic discussions	4 M	386,670	402,573	1,428,090	579,354	30,524	558,878	61,340	74,951
Mean No. of topic reads	5.8 B	1.6 B	712 M	696 M	2.9 B	267 M	2.9 B	227 M	203 M
Common users	*C* _1_	*C* _2_	*C* _3_	*C* _4_	*C* _5_	*C* _6_	*C* _7_	*C* _8_	*C* _9_
Mean No. of retweets	26	9	9	7	8	3	6	8	5
Mean No. of comments	26	13	20	7	13	13	10	16	9
Mean No. of likes	208	127	179	77	133	36	103	120	126
Mean No. of topic discussions	3.7 M	97,982	268,376	925,573	201,429	21,038	139,103	57,291	52,052
Mean No. of topic reads	4.2 B	440 M	460 M	600 M	1.1 B	190 M	750 M	200 M	130 M

*C*_1_ = “Fighting COVID-19”, *C*_2_ = “Novel coronavirus”, *C*_3_ = “Wenliang Li”, *C*_4_ = “Treatment”, *C*_5_ = “Epidemic development”, *C*_6_ = “Epidemic prevention and control”, *C*_7_ = “Confirmed cases”, *C*_8_ = “Isolation”, *C*_9_ = “Support Wuhan”.

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
