# Peer review of "Tracking and Analyzing Public Emotion Evolutions During COVID-19: A Case Study from the Event-Driven Perspective on Microblogs"

_ijerph, 2020, doi:10.3390/ijerph17186888_

Round 1

Reviewer 1 Report

In this work the public's emotional evolutions during COVID-19 in China from social networks and were studied. Further, the authors analyzed the root causes behind these public emotions from the event-driven perspective. A data set was constructed based on microblogs (n=125,672) labeled with COVID-19-related supertopics (n=680) from 40,891 users from Dec 1, 2019 to Feb 17, 2020. Based on the skeleton and key change points of COVID-19 extracted from microblogging contents, the public's emotion evolution modes (accumulated emotions, emotion covariances and emotion transactions) were tracked by phase and further extracted detailed dominant social events. Public emotions exhibited different evolution modes during different phases of COVID-19 evolution. Events about the development of COVID-19 remained hot but generally declined, and the public attention shifted to other aspects of the epidemic, i.e., encouragement, support and treatment. In general, the manuscript is interesting and well organized. There are some minor issues which I encourage the authors to consider:

Minor revisions:

There are several grammatical/style errors. A grammar/style revision has to be carried out before the manuscript can be considered for publication.

Reviewer 2 Report

The manuscript titled tracking and analyzing public emotion evolutions during COVID-19, A Case study from the event-driven perspective on microblogs by Liu et al. is clearly written, but lacks the new physical insight to the public health. The manuscript did a detailed analysis on the public emotions during the different phases of COVID-19 evolution. I  can conclude that the obtained results of emotion evolution are obvious to the present situation ;  Hence I cannot recommend the paper for publication

Reviewer 3 Report

The paper presents an analysis of social network emotion evolution during COVID-19 from microblog activity.

The paper is well organized and readable.

Please, rewrite your paper contributions in the discussion and conclussions section. Contributions are not research work descriptions, but novel proposals and/or results.

In the introduction section, please add a state of the art of microblog and emotions (a current search with both words in the title gets around 40 papers). Interesting things to present are the different types of emotions studied before, evolution has been presented? other type of analysis and how yours is new or why is the best approach. A table with a short summary is desirable.

Reviewer 4 Report

Reviewer comments

General Comments

Some basic info for readers would be helpful – like the definition of a “microblog.”  

How many persons in China use the Sina platform? Is it the only platform available or just the largest platform? What are the age groups of persons that tend to use that platform? I ask because mostly middle aged-persons and older persons in the US use Facebook, while younger people like Instagram or Snapchat better.

For the retweets – can you provide more information about the platform that was used to Tweet? Was is the US-based Twitter or a Chinese-based company?

How did the author account for the use of emojis or emoticons to express emotion? How did you differentiate between sarcasm and true negative emotions?

What objective criteria did you use to define “Large users” and “Small users”? The manuscript said that large users had “users with many fans and readings,” but what does that mean in terms of numbers?

How did you differentiate between actual users that retweeted the blogs, commented, liked, or read the sites from paid bots?

What is the potential role of government censorship in the posting and sharing of comments? I ask because readers like myself may not be familiar with the Chinese government’s policies regarding free speech in social media.

There is so much information in this paper that is great, but the results section needs a lot of work before this paper is ready for publication. Namely, there are quite a few figures that are interesting, but the authors need to reduce the number of figures down to about 5 key figures to illustrate the points.The same with the reporting of the results - just focus on the key messages.

Have you examine how responses from Wuhan differ from surrounding regions?

Minor comments

In Table 1 – it would be helpful to label the datasets according to a description of their origin – something more specific than “Dataset 1”

Results section 3.2 – this section is quite short – please combine with Section 3.1

Figure 1 – what is the Y axis supposed to represent? Please label accordingly.

Figure 3 – Please reword to title to address that the hot events were identifying topics that people were talking about

Figure 3 - Who is Wenliang Li? And how is that person relevant to the pandemic?

Figure 9 – please include a key to explain what the letters lining the axes mean. There is a key but it is not located near the figure in the manuscript

Round 2

Reviewer 4 Report

I think the authors did a great job of addressing my comments.